# Olfactory Enrichment in Hoary Foxes (*Lycalopex vetulus* LUND 1842): A Case Study

**DOI:** 10.3390/ani13091530

**Published:** 2023-05-03

**Authors:** Milene de Paula Figueira, Ita de Oliveira e Silva, Vanner Boere

**Affiliations:** 1Post-Graduate Program at Animal Biology, Universidade Federal de Viçosa, Viçosa 36570-900, MG, Brazil; milene.figueira@gmail.com; 2Post-Graduate Program at Health, Environment and Biodiversity, Universidade Federal do Sul da Bahia, Itabuna 45653-970, BA, Brazil; itabio@hotmail.com; 3Post-Graduate Program at Environmental Sciences and Technology, Universidade Federal do Sul da Bahia, Itabuna 45653-970, BA, Brazil

**Keywords:** animal welfare, environmental enrichment, Canidea, *Lycalopex vetulus*

## Abstract

**Simple Summary:**

Wild animals in captivity need stimuli that increase their well-being. Canids in general have a well-developed sense of smell and are strongly related to environmental stimuli through scent. Therefore, we tested an olfactory enrichment method in five hoary foxes, which was successfully developed in another species of canid. We offered four stimuli (cheese, eggs, meat, and sawdust impregnated with rat urine), and observed the individuals’ reactions that indicated an improvement in well-being before, during, and after exposure to the stimuli. There were no significant changes in behaviors that indicated well-being, although there was no worsening in behaviors suggestive of stress. We suggest that the indifference to stimuli of this little-known species is due to the highly insectivorous diet of the hoary fox.

**Abstract:**

We have tested a method of olfactory environmental enrichment in hoary foxes used in other wild canids in captivity. The individuals were exposed to four olfactory stimuli (meat, mouse urine, cheese, and egg) that were wrapped in cotton bags outside the enclosures at the zoo for five minutes. Behavioral observations were performed using the focal animal method, and all occurrences were recorded. The pre-exposure phase (Basal), exposure phase (Exp), and post-exposure phase and Basal phase (Pos) were analyzed for a period of five minutes. Behavioral responses were categorized as positive, negative, or other. Positive behavior tended to increase (*p* = 0.07) from the Basal phase to the Exp phase, but there was no statistical difference (*p* = 0.31) between the phases. Negative and other behavior did not differ statistically from the Basal phase to the Exp phase (N−, *p* = 0.32; Ot, *p* = 0.35) or Basal to the Pos phase (N−, *p* = 0.18; Ot, *p* = 0.92). The odors used seemed to elicit positive behavior weakly. Negative behavior was stable for the hoary foxes. The method failed to improve the hoary foxes’ welfare. Because their natural diet is based on insects and fruits, it is suggested that the stimuli used in this study have no appetitive value for hoary foxes. The method used with the same olfactory stimuli that were successful in other canid species is unsuitable for hoary foxes.

## 1. Introduction

Captive environments are often monotonous, limited in stimuli, and restrict the performance of behaviors considered normal for the species. Environmental enrichment (EE) is defined as “an improvement in the biological functioning of captive animals resulting from modifications to their environment” [1]. The application of EE must be safe, significant to the individual, and preferably with low administrative costs [1]. As described within the definition itself, EE leads to an increase in animal welfare, which is one of the goals of most zoos [2].

EE is characterized by the introduction of stimuli linked to the social, physical, and sensory contexts of captive animals [3]. Despite knowledge of the high level of olfactory acuity of canids in general [4,5,6,7], studies about olfactory enrichment are little explored. Only 3% of the articles published in the scientific literature deal with olfactory enrichment (OE) in canids [4]. Despite the scarcity of studies on olfactory enrichment for South American canids, recently, some authors have presented a successful and less invasive method for crab-eating fox (*Cerdocyon thous*) [8]. In that experiment, the authors observed 22 crab-eating foxes exposed to four types of food-related odors. Behaviors suggestive of enhanced well-being (“Positive” behaviors) increased. On the contrary, there was a decrement in behaviors considered “negative”, which lowered well-being. These effects remained after withdrawal of stimuli, in the short term [8].

While crab-eating foxes are relatively well studied in captivity, there are few studies on the behavior of hoary foxes (Figure 1). In particular, there are no studies of OE for the hoary fox [7,9]. The hoary fox is a species endemic to the Brazilian Cerrado, being considered “Near Threatened” in the International Union of Conservation Nature extinction risk indices [10]. The greatest threats to the hoary fox are habitat loss, predation by domestic dogs (*Canis lupus familiaris*), and the danger of being run over on the country’s highways [11]. In many cases, animals injured or seized outside of their habitat are taken to recovery centers or zoos, remaining in captivity indefinitely [11].

The diet of foxes is well known to be based largely on insects, particularly Coleoptera [11]. The regular acquisition of insects to feed the foxes in captivity is not feasible because it would require an infrastructure that demands high costs. For this reason, a mix of dog food, meat, and some vegetables is regularly offered in the diet of zoo canids, including foxes. Little is known about the social interactions of foxes, appearing to be restricted to the pair’s interaction during the mating season and the mother’s relationship with her young [11]. In a literature review, we did not find systematic studies on the relationships of foxes in captivity when housed in pairs or with more animals. With this scenario, we deduce that the captive environment is not stimulating, consequently reducing the possibility of satisfying the foxes’ behavioral needs.

Based on studies on environmental enrichment in canids [4,8], we hypothesized that the introduction of different, non-noxious olfactory stimuli could increase the well-being of captive hoary foxes, as observed in another study on crab-eating foxes [8]. Because of the lack of knowledge on how olfactory stimuli (OS) can be introduced in an EE program for rarely studied canid species, the current study sought to investigate the behavioral response of hoary foxes exposed to different odors in captivity. The ultimate goals of olfactory stimuli are responses with exploratory behaviors, play, non-agonistic interactions, and relaxation; when these behaviors increase, we interpret that there is an increase in the well-being of the foxes.

## 2. Materials and Methods

The study was carried out with five captive hoary foxes (Figure 1) in the Ecological Zoo Park of São Carlos (PESC), in São Carlos, SP, Brazil. The individuals were adults (one male and four females) between two and eight years old. The foxes were housed in a pair and a trio (2 females and a male), in enclosures with an area of approximately 100 m^2^, surrounded by wire fences on three sides and a wall in the back. Inside the enclosure, there was a shelter for the foxes to hide and rest. Tree trunks, a bush, and natural stones also structured the exhibit. At the back of the enclosure, there was an indoor area with bowls for drinking water and eating. The foxes were fed a mix of fresh fruits, protein of animal origin, and industrial dog food in the morning. The foxes were healthy, and neither pregnant females nor puppies were present during the study.

The OE method tested in hoary foxes in the present investigation was adapted from the study by Figueira et al. (2021) [8] on crab-eating foxes. Due to the absence of studies in the scientific literature on OS for hoary foxes, the odorous stimuli were adapted from the study on crab-eating foxes [8]. The OS were 100 g of fresh minced beef; 100 g of chopped parmesan cheese; two boiled and chopped chicken eggs; and approximately 100 g of sawdust removed from boxes containing rodents. A detailed ethogram for captive hoary foxes was not found in the scientific literature to determine behavioral welfare. For this reason, an ethogram (Table 1) was developed as an adaptation from the description of captive crab-eating fox behavior [8]. In order to better analyze the effects of OS, behavior was categorized as positive (P+), negative (N−), and other (Ot). Based on the scientific literature, the P+ category contains behaviors that increase an animal’s welfare, while the N− category contains undesirable signs of distress [12,13]. The Ot behaviors were considered ambiguous or indifferent to OS and consequently do not influence the welfare of the hoary foxes.

The olfactory stimuli were placed inside permeable cotton bags, which allowed the animals to sense their odors without being able to see them. All of the bags were the same color and size and were washed with mild soap after each use. The OS were positioned in front of and outside each enclosure. The observation sessions were in the morning, before food was placed for the foxes and without visitors in the zoo.

The behavior of the individuals was recorded with digital cameras (Samsung^®^ ST77, Daegu, Republic of Korea), which were mounted on tripods in front of the cage at a height of 1.5 m. The filming took place between 8 am and 10 am, before the feeding of the animals by zoo staff.

The filming sessions of the foxes in each enclosure lasted one morning, on different days (Figure 2). On the day, four OS sessions were conducted, one for each attractive stimulus. The order in which the OS were presented had been previously defined by chance. Each session lasted five minutes, with one-minute intervals between sessions. After positioning the camera, we filmed for 5 min without exposing the subjects to any stimuli. We called this phase “warm up”, so that the individuals would get used to the presence of the film camera and movements of the researcher. The approach and movement of the researcher could scare the hoary foxes. The warmup session serves to not suddenly scare the animals. Soon after, the Basal session began; this was a five-minute session where the animal was filmed without presentation of any OS. Following that, the exposure session (Exp) began when the researcher placed the OS in front of the enclosure and left again. At the end of the Exp session, the researcher entered in front of the enclosure to remove the OS and again left. This marked the beginning of the post-exposure (Pos) session, where the individual was filmed for five more minutes without the stimulus. After that, the session ended.

The behavioral responses of the individuals were collected following the focal animal method and recording all behaviors [14]. The total time of each behavior was counted with the aid of the Prostcom behavioral analysis software [15]. During the first minute of each phase (Basal, Exp, and Pos), while the researcher was in front of the animal’s field of view, the behaviors were not recorded in the software. The means of behavioral responses (P+, N−, and Ot) from the five hoary foxes were calculated from the recorded time for each set of stimuli sessions. All comparisons were performed in order to verify whether there were any changes in behavior in the Exp and Pos phases in relation to the Basal session. When changes in behavior occurred, it was assessed whether they increased or decreased and whether they remained after the olfactory stimulus was removed. Due to the small sample size, which distorts to a non-normal distribution of the data, non-parametric analyses were performed, applying the Wilcoxon test for paired samples [16]. All statistical tests followed a two-tailed distribution, with an alpha level < 5%.

## 3. Results and Discussion

The behavior duration (s) of the P+ category tended to increase (*p* = 0.07) from the Basal phase to the Exp phase, but it was not statistically different (*p* = 0.31) in the Pos phase compared to the Basal phase (Table 2). The average duration for the N− and Ot categories did not differ significantly (N−, *p* = 0.32; Ot, *p* = 0.35) from the Basal phase to the Exp phase or from the Pos phase to the Basal phase (N−, *p* = 0.18; Ot, *p* = 0.92; Table 2).

Canids use smell as one of their principal means of communication and exploration of the environment [9], but in this experiment, the olfactory stimuli were not able to alter behavior significantly. The hoary foxes seemed to be indifferent to the olfactory stimuli with the method used. The lack of differences in N− and Ot, together with the weak effect on P+ between phases, strongly suggests that the response of hoary foxes to olfactory stimuli was one of indifference.

Odors are a complex mixture of several volatile compounds, whose composition is dependent on concentration and the chemical family of the molecules [17]. Some edible items share common volatile compounds, having odor-like organoleptic characteristics and making them appetizing for an animal species. The stimuli used (meat, egg, mouse urine, and Parmesan cheese) have volatile components that differ in their composition [18,19,20,21], and are items that are not listed in the scientific literature as food ingested by hoary foxes. The hoary fox is the most specialized South American canid [11], with a diet largely based on termites, coleopterans, and fruits [22], having insectivore dentition [23]. Insects and fruits predominate in the hoary fox diet, whose compositions of volatile molecules, such as a high concentration of alkaloids in insects, are apparently different from the OS that were utilized in the experiment [24]. The stimuli to which the hoary foxes were exposed may therefore not have had enough appetitive value to elicit a behavioral response indicative of either increased or decreased welfare.

Despite the tendency for positive behaviors to increase during exposure to the stimulus, overall, the mean time spent on these activities was low compared to the category of other behaviors. The mean duration of negative behaviors was also low compared to the category of other behaviors. Looking from another perspective, the average times of the positive or negative behavior occupied a fraction of the session’s time, while the average time of other behaviors occupied between 98.8% (Phase Exp) to 99.9% (Phase Pos) of the time of the sessions. The very low manifestation of positive and negative behaviors suggests that the hoary foxes were indifferent to the stimuli in our method adapted from the schedule proposed by Figueira and collaborators [8] for crab-eating foxes. In that experiment, positive behaviors increased during exposure and remained high after stimulus withdrawal.

Environmental enrichment using olfactory cues that simulate food must have an appetitive value for individuals, in order to encourage behaviors from their natural behavioral repertory [1]. The stimuli used in this experiment do not appear to have been reported in the scientific literature for free-ranging hoary foxes, and may not have biological significance to motivate individuals to increase positive behaviors. This study suggests that it is necessary to know aspects of the feeding ecology of each species to expose captive animals to olfactory enrichment.

The sample size may have been insufficient to demonstrate the effect of stimuli on positive behaviors, given the statistical trend found between the Basal phase and the Exp phase. The opposite might also be true, that is, an increased sample size might clearly show a lack of significant difference between the phases. Therefore, the results based on a sample of five individuals do not allow the conclusion that the method using the four stimuli is appropriate to increase the well-being of hoary foxes. Although we did not obtain results that indicated an increase in the welfare of hoary foxes, the reporting of the results is recommended for practical and ethical reasons. Results that frustrated expectations or hypotheses are less publicized than results that prove the expectations of the researchers, which can lead to biased conclusions [25]. Resources can be saved by avoiding procedures that do not appear to be adequate for improving animal welfare. Despite the inconclusive results for the environmental enrichment, this behavioral investigation is original, since it is the first study to evaluate olfactory stimuli in hoary foxes.

## 4. Conclusions

The olfactory environmental enrichment method used in other canid species did not seem suitable for hoary foxes. Due to the highly specialized food biology of hoary foxes, the olfactory stimuli used do not seem to be attractive enough to modify the behavior that indicates an improvement in well-being. Other more appropriate stimuli (e.g., insect odor) could lead to the success of environmental enrichment using the method described in this study. Finally, it is necessary to consider the ecology of each animal species to introduce environmental enrichment to captive individuals.

## Figures and Tables

**Figure 1 animals-13-01530-f001:**
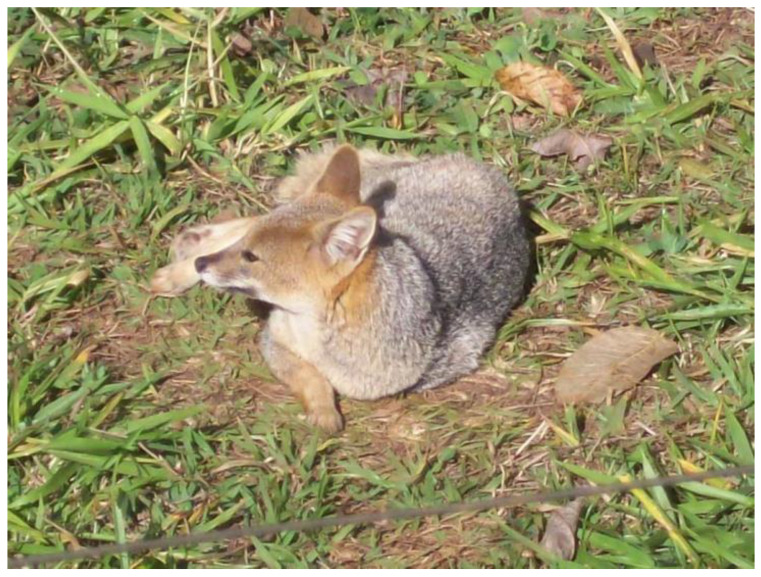
A male adult hoary fox (*Lycalopex vetulus*) from Parque Ecológico de São Carlos.

**Figure 2 animals-13-01530-f002:**
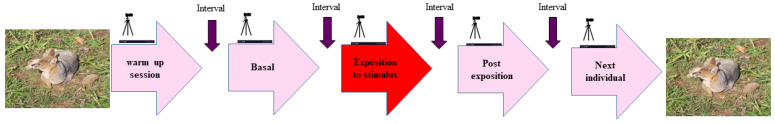
Scheme of observations with the flow and duration of behavior of foxes exposed to olfactory stimuli.

**Table 1 animals-13-01530-t001:** An ethogram for the study of olfactory enrichment (OE) for the hoary fox (*Lycalopex vetulus*). P+, positive behavior; N−, negative behavior; Ot, other behavior.

Behavior	Behavior Description	Category
Play	Individual interacts with the environment or with another animal in a relaxed way.	P+
Non-agonistic social interaction	Individual shows non-agonistic behavior and seeks contact with another animal such as licking or grooming.	P+
Attempting to reach the OE	Individual tries to reach the OE with its paw through the cage.	P+
Self-maintenance	Individual bites or licks, slowly and calmly, parts of its own body.	P+
Sniffing	Individual moves its nostrils, pointing towards objects or regions of the enclosure.	P+
Sniff or point OE	Animal points its snout in the direction where the OE is or was placed.	P+
Agonistic behavior	Individual shows signs of aggression such as growls, baring of teeth, scratching, or biting another animal.	N−
Biting the cage	Individual bites or pulls the cage with its teeth.	N−
Yawning	To open the mouth with an apparently deep inhalation and sighing or heavy exhalation; usually the individual shows his teeth, closes his eyes, and extends his neck forward. This whole procedure takes one to three seconds, approximately.	N−
Scratching itself	Individual rubs one leg or its mouth vigorously on its skin or hair.	N−
Stereotypy	Individual performs repeated movements more than three times for no apparent reason.	N−
Climbing the railing or wall	Individual stands up and supports its front limbs on the railings or walls of the enclosure.	N−
Sneeze	Self-defined behavior.	N−
Others	This is any activity not listed in the behaviors described as P+ or N−	Ot
Out of sight	Focal animal is out of sight for the observer.	Ot

**Table 2 animals-13-01530-t002:** Behavioral responses of the hoary fox (*Lycalopex vetulus*) to olfactory stimuli (OS) during the Basal (no stimuli), Exp (exposed to stimuli), and Pos (after stimuli) phases. Wilcoxon test (Z) and significance level (P) in the comparison of the average time spent for different behavioral responses: P+ (positive behaviors), N− (negative behaviors), and Ot (other behavior). n = 5.

Bahavioral Category	OS	Mean ± Standard Error (s)	*Z* Value	*p*
P+	Basal	0.65 ± 0.65	−1.83	0.07 *
	Exp	3.51 ± 1.88		
N−	Basal	0.15 ± 0.15	−1.00	0.32
	Exp	0.00		
Ot	Basal	299.21 ±0.80	−0.94	0.35
	Exp	296.50 ± 1.87		
P+	Basal	0.65 ±0.65	−1.00	0.31
	Pos	0.00		
N−	Basal	0.15 ±0.15	−1.34	0.18
	Pos	0.41 ± 0.28		
Ot	Basal	299.21 ± 0.80	-0.11	0.92
	Pos	299.60 ± 0.28		

* Tendency to significant differences (*p* interval = 0.05 to 0.075) [16].

## Data Availability

Authors will send the data under request.

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
