# Peer review of "Olfactory Enrichment in Hoary Foxes (*Lycalopex vetulus* LUND 1842): A Case Study"

_animals, 2023, doi:10.3390/ani13091530_

Round 1
Reviewer 1 Report
In the manuscript, the authors response the difficult problem of scent stimulation of Brazilian foxes kept in the zoo.
Nevertheless, the attractants were not selected based on the behavior of the species, but on the basis of existing knowledge for the canid family. The authors point to this as the main conclusion of the study. It would be worth preparing at least one of the scent stimuli especially for the studied fox species.
Nevertheless, the study prepared correctly, the literature selected appropriate. I recommend enriching the discussion with current sources relating to the behavioral activity of foxes based on olfactory stimulation and justifying the broader justification of the research method.
The value of the work will be increased by attaching a diagram or photograph depicting the locations of cotton bags with olfactory.
Photograph 1 of the fox could show the animal in a standing position, allowing the reader to get acquainted visually with the species described.
The work requires corrections, particular expansion of the content in the discussion. After making changes, I recommend the manuscript for printing.
Author Response
April 7th. 2023
Responses to Reviewer 1
Dear Reviewer
We appreciate the patience and effort made by the reviewer on our manuscript. We appreciate your suggestions, which were invaluable to increase the quality of the text. We agreed with all of the reviewers' suggestions. In some manuscript arguments where we had some differences from the reviewer, we made a long argument to explain our point of view. Overall, we made several modifications to the manuscript that greatly improved the quality, thanks to the reviewer's comments. Below, we respond to each reviewer's comment.
Reviewer’s commentary:
“In the manuscript, the author's response is the difficult problem of scent stimulation of Brazilian foxes kept in the zoo.
Nevertheless, the attractants were not selected based on the behavior of the species, but on the basis of existing knowledge for the canid family. The authors point to this as the main conclusion of the study. It would be worth preparing at least one of the scent stimuli especially for the studied fox species."
Response: Yes, we tested the method used by our research group which was successful in crab-eating foxes (Figueira et al., 2021) and maned-wolves (in press, Figueira et al., 2023). In the current study with Hoary foxes, our results were absolutely not indicative of increased welfare, because only positive behaviors increased; but did not decrease the negative behaviors. This result suggests that the stimuli were not neutral and did not cause aversion, but they were not strongly attractive.
Reviewer’s commentary:
Nevertheless, the study prepared correctly, the literature selected appropriate. I recommend enriching the discussion with current sources relating to the behavioral activity of foxes based on olfactory stimulation and justifying the broader justification of the research method.
Response: We would like to see studies on environmental enrichment for Hoary foxes, particularly with olfactory stimuli. To our knowledge, there are no studies. Therefore, we believe that this is the first study of olfactory enrichment in Hoary foxes.
Reviewer’s commentary:
The value of the work will be increased by attaching a diagram or photograph depicting the locations of cotton bags with olfactory.
Response: We insert a figure (Fig. 2) with the scheme of observations.
Reviewer’s commentary:
Photograph 1 of the fox could show the animal in a standing position, allowing the reader to get acquainted visually with the species described.
Response: This is an interesting observation. We selected an original photograph of a fox in a resting position, but with its snout apparently pointed at something, a stimulus. This photograph has a semiotic value for "meaning" that there was not enough attraction for the olfactory stimuli, which elicited exploratory motor behavior from the foxes.
Reviewer’s commentary:
The work requires corrections, particular expansion of the content in the discussion. After making changes, I recommend the manuscript for printing.
Response: We corrected some sentences, which was color-marked in the text.
REFERENCES
Figueira, M P, Silva, FFR, Ribeiro, A, Silva, IO, Boere, V. The behavioral response of the crab-eating fox to olfactory en-richment. Applied Animal Behavior Science. 2021, Sep; 242, 105430, doi: 10.1016/j.applanim.2021.105430
Figueira, M P, Silva, FFR, Ribeiro, A, Silva, IO, Boere, V. Behavioral responses of captive Maned Wolves to olfactory enrichment: A preliminary study. Revista Argentina de Ciencias del Comportamiento. 2023, 65 In press.

Reviewer 2 Report
This is a very small study (both in terms of the number of animals and also the length and frequency of observations) that examined the effects of olfactory enrichment on the behavior and welfare of Hoary foxes.
This form of environmental enrichment is often underutilized, and I salute the authors for exploring this avenue. In addition to the small sample size, there are some additional issues relating to the methodology (particularly, related to the presentation of the stimuli, and the data collection methods), no specific information about the 5 individuals that were observed (e.g., exact age, time spent together, dominance), and no information about the order of stimuli presentation. The Results and Discussion section needs to be improved as well. The units of measurement are missing, there is no discussion about the social interactions among the animals that may affect the behavioral outcome, and wind conditions during test trials. Finally, you stated that the tests were conducted in the morning before food was provided. This was probably a deviation from the norm in which the animals probably got their food as soon as they were given access to the exhibition. Is that correct? If it is, what implications would that have on the animals’ behavior?
My specific comments are listed below:
Simple Summary
16: …that indicated an improvement in well-being.
Introduction
44-45: this sentence is unclear.
49-53: Please rephrase this paragraph. It reads a bit awkward. Also, please specify the “positive” and “negative” behaviors.
Materials and Methods
69-70: I find “habituated to captivity” to be somewhat too general. Which aspects of captivity were the animals habituated to and what criteria were employed to determine that the animals became habituated? As I assume that those aspects were not examined, perhaps you can simply state that the animals were either born or arrived at a young age to PESC (and provide a table with this and additional information).
Table 1: Please avoid using the behavioral term in the behavior description (e.g., Play/playful). Please only use objective terms in the description of the behavior (e.g., do not use ‘friendly”), and provide a better description of “yawning”.
88: do P+ behaviors increase welfare or reflect a positive welfare state (or both)?
93-94: what was the distance between the enclosure and the stimuli? Did you consider the intensity and direction of the wind during the observations? A diagram of the enclosures would be useful.
106-109: please rewrite this section.
108: what did you base your assumption on regarding habituation?
XX: please add the order of stimuli presentation for each group.
116-117: can you please clarify if you observed all the animals in a group during a 5-minute observation or one animal at a time? It sounds like you may have observed the entire group during each observation, and recorded data for each individual. In that case, the sampling rule would be ad-lib sampling or behavior sampling (depending on whether or not the information was recorded per individual), and the recording rule would be continuous recording.
120: you wrote “averages” – was it the mean, median, or mode?
122-128, 191: I suggest changing the word “verified” to “assessed”, “evaluated”, or “tested”.
Results and Discussion
130-134: units of measurement are missing.
134-137: please delete.
147:156: this begs the question of why did you choose stimuli that were not ecologically relevant?
164-176: the terminology of using positive and negative is confusing in this context since you have used it before to categorize the behaviors. Please change to a different term.
200: rewarding?
Author Response
April 7th. 2023
Response to Reviewer 2
Dear Reviewer
We appreciate the patience and effort made by the reviewer on our manuscript. We appreciate your suggestions, which were invaluable to increase the quality of the text. We agreed with all of the reviewers' suggestions. In some manuscript arguments where we had some differences from the reviewer, we made a long argument to explain our point of view. Overall, we made several modifications to the manuscript that greatly improved the quality, thanks to the reviewer's comments. Below, we respond to each reviewer's comment.
Reviewer’s commentary:
Comments and Suggestions for Authors
This is a very small study (both in terms of the number of animals and also the length and frequency of observations) that examined the effects of olfactory enrichment on the behavior and welfare of Hoary foxes.
This form of environmental enrichment is often underutilized, and I salute the authors for exploring this avenue. In addition to the small sample size, there are some additional issues relating to the methodology (particularly, related to the presentation of the stimuli, and the data collection methods), no specific information about the 5 individuals that were observed (e.g., exact age, time spent together, dominance), and no information about the order of stimuli presentation.
Response: Yes, the sample is small. The population of Hoary foxes that can be observed in captivity is small, even in Brazil, where the geographical distribution of the species is. Therefore, there is a restriction on a larger sample.
In line 68, we describe some characteristics of individuals, thus described: “The individuals were adults (one male and four females) between two and eight years old, habituated to captivity because they were born at the institution or arrived very young. The foxes were housed in a pair and a trio (2 females and a male)…”. We know of no studies on dominance in Hoary foxes.
Reviewer’s commentary:
The Results and Discussion section needs to be improved as well. The units of measurement are missing, there is no discussion about the social interactions among the animals that may affect the behavioral outcome, and wind conditions during test trials. Finally, you stated that the tests were conducted in the morning before food was provided. This was probably a deviation from the norm in which the animals probably got their food as soon as they were given access to the exhibition. Is that correct? If it is, what implications would that have on the animals’ behavior?
Response: Although wind strength and direction can influence the foxes' perception of odors, we selected observation days with good weather, no rain, and no strong winds. Furthermore, the animals' enclosures are not open fields; the enclosures are in a complex of buildings and other zoo aviaries, which serve as a windbreak.
Hoary foxes have a zoo routine. Meals are provided at 10:00 a.m. It is well known that animals learn many zoo routines, including feeding times (Clark, 2019). Our procedures were before offering food. We did not expose the animals to other stimuli (visual, auditory, etc.), with the exception of odors. Canids relate to the environment a lot through smell, so whatever the stimulus. The stimuli offered are different from the routine of the animals. Thus, a reaction would be expected. Therefore, there does not seem to have been a connection between the routine schedule of offering the zoo's food and the period in which we carried out the experiment.
Hoary foxes are little known for their social interactions. It is known that they have little sociable habits, generally more observable in the interactions between couples during the mating period, and in the interaction between the mother and the cubs. We observed an apparent indifference between the animal foxes in each enclosure (Personal observation). In many studies on environmental enrichment behavior, the observations are from enclosures with more than one animal, but the effects have not been invalidated, even when the authors do not describe the social interactions of individuals within the enclosure (Vidal et al., 2016; Regaiolli et al., 2020). Furthermore, when routine environmental enrichment is carried out in zoos, it is generally avoided that animals in the same enclosure are separated. Our experimental design is more “realistic”. Therefore, we do not observe that it is relevant to describe social interactions.
Reviewer’s commentary:
My specific comments are listed below:
Simple Summary
16: …that indicated an improvement in well-being.
Response: Done.
Reviewer’s commentary:
Introduction
44-45: this sentence is unclear.
49-53: Please rephrase this paragraph. It reads a bit awkward. Also, please specify the “positive” and “negative” behaviors.
Response: We modified the paragraph, including explaining what is "positive behavior" and "negative behavior", according to the authors of the reference [8]
Reviewer’s commentary:
Materials and Methods
69-70: I find “habituated to captivity” to be somewhat too general. Which aspects of captivity were the animals habituated to and what criteria were employed to determine that the animals became habituated? As I assume that those aspects were not examined, perhaps you can simply state that the animals were either born or arrived at a young age to PESC (and provide a table with this and additional information).
Response: As stated above, animals learn zoo routines (Clark, 2019) in order to mitigate the fear behaviors observed in naïve animals. To avoid misunderstandings, we removed the term.
Table 1: Please avoid using the behavioral term in the behavior description (e.g., Play/playful). Please only use objective terms in the description of the behavior (e.g., do not use ‘friendly”), and provide a better description of “yawning”.
Response: Done.
88: do P+ behaviors increase welfare or reflect a positive welfare state (or both)?
Response: We defined that “P+ category contains behaviors that increase an animal’s welfare”.
93-94: what was the distance between the enclosure and the stimuli? Did you consider the intensity and direction of the wind during the observations? A diagram of the enclosures would be useful.
Response: the distance between the enclosure and the stimuli is 1 m. About the wind and the diagram, we already answered previously.
106-109: please rewrite this section.
Response: We rewrote the sentence.
108: what did you base your assumption on regarding habituation?
Response: We explain in the text of the manuscript what "habituation" means.
XX: please add the order of stimuli presentation for each group.
Response: The order of stimuli presentation was random.
116-117: can you please clarify if you observed all the animals in a group during a 5-minute observation or one animal at a time? It sounds like you may have observed the entire group during each observation, and recorded data for each individual. In that case, the sampling rule would be ad-lib sampling or behavior sampling (depending on whether or not the information was recorded per individual), and the recording rule would be continuous recording.
Response: We observed one animal at a time.
120: you wrote “averages” – was it the mean, median, or mode?
Response: We changed the word.
122-128, 191: I suggest changing the word “verified” to “assessed”, “evaluated”, or “tested”.
Response: Done.
Reviewer’s commentary:
Results and Discussion
130-134: units of measurement are missing.
Response: We insert the unit (seconds).
134-137: please delete.
Response: Done.
147:156: this begs the question of why did you choose stimuli that were not ecologically relevant?
Response: With the results obtained by our group with Crab-eating foxes and Maned Wolves, referred to above, we hypothesized that the method could be applied to Hoary foxes. We consider that the long period of captivity and the diet based on dog food + meat given to the foxes could have modified the food preference. We discuss that insect-based food specialization (Coleopterans etc.) has a greater effect than we would expect as a stimulus to hoary foxes.
164-176: the terminology of using positive and negative is confusing in this context since you have used it before to categorize the behaviors. Please change to a different term.
Response: We changed the sentences.
200: rewarding?
Response: We deleted the word.
REFERENCES
Clark, F. (2020). What Is There to Learn in a Zoo Setting?. Zoo Animal Learning and Training, 83-100.
Regaiolli, B., Rizzo, A., Ottolini, G., Miletto Petrazzini, M. E., Spiezio, C., & Agrillo, C. (2019). Motion illusions as environmental enrichment for zoo animals: A preliminary investigation on lions (Panthera leo). Frontiers in Psychology, 10, 2220.
Vidal, L. S., Guilherme, F. R., Silva, V. F., Faccio, M. C. S. R., Martins, M. M., & Briani, D. C. (2016). The effect of visitor number and spice provisioning in pacing expression by jaguars evaluated through a case study. Brazilian Journal of Biology, 76, 506-510.

Round 2
Reviewer 2 Report
Dear Reviewer
We appreciate the patience and effort made by the reviewer on our manuscript. We appreciate your suggestions, which were invaluable to increase the quality of the text. We agreed with all of the reviewers' suggestions. In some manuscript arguments where we had some differences from the reviewer, we made a long argument to explain our point of view. Overall, we made several modifications to the manuscript that greatly improved the quality, thanks to the reviewer's comments. Below, we respond to each reviewer's comment.
Reviewer’s commentary:
Comments and Suggestions for Authors
This is a very small study (both in terms of the number of animals and also the length and frequency of observations) that examined the effects of olfactory enrichment on the behavior and welfare of Hoary foxes.
This form of environmental enrichment is often underutilized, and I salute the authors for exploring this avenue. In addition to the small sample size, there are some additional issues relating to the methodology (particularly, related to the presentation of the stimuli, and the data collection methods), no specific information about the 5 individuals that were observed (e.g., exact age, time spent together, dominance), and no information about the order of stimuli presentation.
Response: Yes, the sample is small. The population of Hoary foxes that can be observed in captivity is small, even in Brazil, where the geographical distribution of the species is. Therefore, there is a restriction on a larger sample.
In line 68, we describe some characteristics of individuals, thus described: “The individuals were adults (one male and four females) between two and eight years old, habituated to captivity because they were born at the institution or arrived very young. The foxes were housed in a pair and a trio (2 females and a male)…”. We know of no studies on dominance in Hoary foxes.
- Thank you for your reply. I still think that it would be beneficial to have a table with the information per animal (rather than the range of ages). This is a solitary species; how does being housed in a social setting might play a part in the behavior observed here?
Reviewer’s commentary:
The Results and Discussion section needs to be improved as well. The units of measurement are missing, there is no discussion about the social interactions among the animals that may affect the behavioral outcome, and wind conditions during test trials. Finally, you stated that the tests were conducted in the morning before food was provided. This was probably a deviation from the norm in which the animals probably got their food as soon as they were given access to the exhibition. Is that correct? If it is, what implications would that have on the animals’ behavior?
Response: Although wind strength and direction can influence the foxes' perception of odors, we selected observation days with good weather, no rain, and no strong winds. Furthermore, the animals' enclosures are not open fields; the enclosures are in a complex of buildings and other zoo aviaries, which serve as a windbreak.
- Please add this information to the manuscript. This is a very basic point that has to be mentioned. You are attempting to assess the effect of stimuli on behavior – reading your paper, it isn’t clear if the animals were even exposed to the stimuli due to wind.
Hoary foxes have a zoo routine. Meals are provided at 10:00 a.m. It is well known that animals learn many zoo routines, including feeding times (Clark, 2019). Our procedures were before offering food. We did not expose the animals to other stimuli (visual, auditory, etc.), with the exception of odors. Canids relate to the environment a lot through smell, so whatever the stimulus. The stimuli offered are different from the routine of the animals. Thus, a reaction would be expected. Therefore, there does not seem to have been a connection between the routine schedule of offering the zoo's food and the period in which we carried out the experiment.
Hoary foxes are little known for their social interactions. It is known that they have little sociable habits, generally more observable in the interactions between couples during the mating period, and in the interaction between the mother and the cubs.
- Precisely my point. Their current social environment is different than the one employed in the wild. I think that it is important to discuss the results in this context.
We observed an apparent indifference between the animal foxes in each enclosure (Personal observation). In many studies on environmental enrichment behavior, the observations are from enclosures with more than one animal, but the effects have not been invalidated, even when the authors do not describe the social interactions of individuals within the enclosure (Vidal et al., 2016; Regaiolli et al., 2020). Furthermore, when routine environmental enrichment is carried out in zoos, it is generally avoided that animals in the same enclosure are separated. Our experimental design is more “realistic”. Therefore, we do not observe that it is relevant to describe social interactions.
- Obviously, observations are frequently done in a group setting, but that was not the issue. The issue raised here is that the results have to be discussed in the appropriate context. The social environment and potential social interactions may have an impact on the outcome variable. I think that this is a critical part that is currently missing from the manuscript.
Reviewer’s commentary:
My specific comments are listed below:
Simple Summary
16: …that indicated an improvement in well-being.
Response: Done.
Reviewer’s commentary:
Introduction
44-45: this sentence is unclear.
49-53: Please rephrase this paragraph. It reads a bit awkward. Also, please specify the “positive” and “negative” behaviors.
Response: We modified the paragraph, including explaining what is "positive behavior" and "negative behavior", according to the authors of the reference [8]
Reviewer’s commentary:
Materials and Methods
69-70: I find “habituated to captivity” to be somewhat too general. Which aspects of captivity were the animals habituated to and what criteria were employed to determine that the animals became habituated? As I assume that those aspects were not examined, perhaps you can simply state that the animals were either born or arrived at a young age to PESC (and provide a table with this and additional information).
Response: As stated above, animals learn zoo routines (Clark, 2019) in order to mitigate the fear behaviors observed in naïve animals. To avoid misunderstandings, we removed the term.
- The term is still there- The study was carried out with five captive hoary foxes (Figure 1) in the Ecological Zoo Park of São Carlos (PESC), in São Carlos, SP, Brazil. The individuals were adults (one male and four females) between two and eight years old, habituated to captivity because they were born at the institution or arrived very young.
Table 1: Please avoid using the behavioral term in the behavior description (e.g., Play/playful). Please only use objective terms in the description of the behavior (e.g., do not use ‘friendly”), and provide a better description of “yawning”.
Response: Done.
88: do P+ behaviors increase welfare or reflect a positive welfare state (or both)?
Response: We defined that “P+ category contains behaviors that increase an animal’s welfare”.
93-94: what was the distance between the enclosure and the stimuli? Did you consider the intensity and direction of the wind during the observations? A diagram of the enclosures would be useful.
Response: the distance between the enclosure and the stimuli is 1 m. About the wind and the diagram, we already answered previously.
- I didn’t see that this was added to the manuscript. How can others replicate this study without having the specific methodology? Please add the information and a diagram.
106-109: please rewrite this section.
Response: We rewrote the sentence.
108: what did you base your assumption on regarding habituation?
Response: We explain in the text of the manuscript what "habituation" means.
- The “explanation” you provided is lacking at best. Habituation is a form of non-associative learning, and in order for one to state that an animal was habituated to a specific stimulus, there has to be a quantified assessment that showed the behavioral change. Otherwise, you cannot use this term. At best, you can say that the “warm-up” sessions were done to decrease the animals’ responses to the camera and observer.
XX: please add the order of stimuli presentation for each group.
Response: The order of stimuli presentation was random.
- I understand that the order was determined randomly, but what was it in practice? Were both groups presented in the same order? Again, it is difficult to interpret the results without it.
116-117: can you please clarify if you observed all the animals in a group during a 5-minute observation or one animal at a time? It sounds like you may have observed the entire group during each observation, and recorded data for each individual. In that case, the sampling rule would be ad-lib sampling or behavior sampling (depending on whether or not the information was recorded per individual), and the recording rule would be continuous recording.
Response: We observed one animal at a time.
- Still unclear. Do you mean that each stimulus was present for 5 minutes X the number of individuals in the group (i.e., 2 or 3)?
120: you wrote “averages” – was it the mean, median, or mode?
Response: We changed the word.
122-128, 191: I suggest changing the word “verified” to “assessed”, “evaluated”, or “tested”.
Response: Done.
Reviewer’s commentary:
Results and Discussion
130-134: units of measurement are missing.
Response: We insert the unit (seconds).
134-137: please delete.
Response: Done.
147:156: this begs the question of why did you choose stimuli that were not ecologically relevant?
Response: With the results obtained by our group with Crab-eating foxes and Maned Wolves, referred to above, we hypothesized that the method could be applied to Hoary foxes. We consider that the long period of captivity and the diet based on dog food + meat given to the foxes could have modified the food preference. We discuss that insect-based food specialization (Coleopterans etc.) has a greater effect than we would expect as a stimulus to hoary foxes.
164-176: the terminology of using positive and negative is confusing in this context since you have used it before to categorize the behaviors. Please change to a different term.
Response: We changed the sentences.
- Frustrated the predictions…? I suggest a better wording.
200: rewarding?
Response: We deleted the word.
REFERENCES
Clark, F. (2020). What Is There to Learn in a Zoo Setting?. Zoo Animal Learning and Training, 83-100.
Regaiolli, B., Rizzo, A., Ottolini, G., Miletto Petrazzini, M. E., Spiezio, C., & Agrillo, C. (2019). Motion illusions as environmental enrichment for zoo animals: A preliminary investigation on lions (Panthera leo). Frontiers in Psychology, 10, 2220.
Vidal, L. S., Guilherme, F. R., Silva, V. F., Faccio, M. C. S. R., Martins, M. M., & Briani, D. C. (2016). The effect of visitor number and spice provisioning in pacing expression by jaguars evaluated through a case study. Brazilian Journal of Biology, 76, 506-510.
Author Response
April 9th. 2023
Response to reviewer 2 - 2nd round
Dear Reviewer
We welcome your comments and suggestions. We made substantial changes in the first round of this manuscript review, accepting all suggestions. In the current evaluation, we agree with some points of view, but we have some explanations about our work.
Below, we present our arguments in detail:
Reviewer’s commentary: The term is still there-
Response: We deleted the term.
Reviewer’s commentary: I didn’t see that this was added to the manuscript. How can others replicate this study without having the specific methodology? Please add the information and a diagram.
Response: The general test scheme and observations were written in the manuscript. This is not a laboratory study with all variables controlled. If one replicates the study, one must do so within the conditions in which the animals live, their enclosures, surrounding buildings, trees, climate, zo routines, etc. Therefore, we kindly disagree to make a specific diagram and specify the weather conditions.
Reviewer’s commentary: The “explanation” you provided is lacking at best. Habituation is a form of non-associative learning, and in order for one to state that an animal was habituated to a specific stimulus, there has to be a quantified assessment that showed the behavioral change. Otherwise, you cannot use this term. At best, you can say that the “warm-up” sessions were done to decrease the animals’ responses to the camera and observer.
Response: We deleted the term.
Reviewer’s commentary: I understand that the order was determined randomly, but what was it in practice? Were both groups presented in the same order? Again, it is difficult to interpret the results without it.
Response: The word “randomly” in Statistics, is “in a way that gives each item in a set the equal probability of being chosen”. Thus, the sequence of individuals and stimuli was random.
Reviewer’s commentary: Still unclear. Do you mean that each stimulus was present for 5 minutes X the number of individuals in the group (i.e., 2 or 3)?
Response: Each individual was observed only once, being exposed to the four stimuli in a randomly determined sequence. In our point of view, it is explicit in the text, with the following sentences: “The shooting sessions of the foxes in each enclosure lasted one morning, on different days (Figure 2). On the day, four OS sessions were conducted, one for each attractive stimulus. The order in which the OS were presented had been previously defined by chance. Each session lasted five minutes, with one-minute intervals between sessions. After positioning the camera, we filmed for 5 minutes, but without exposing the subjects to any stimuli” (Lines 102-107).
Reviewer’s commentary: Frustrated the predictions…? I suggest a better wording.
Response: We have modified the sentence.

Round 3
Reviewer 2 Report
-
Author Response
Response to Editor Comments
Point 1: Dear Authors
thank you for sublitting this manuscript.
However I encourage you to improve the introduction and the discussion by adding more scientific papers. This would help understanding why you conducted this study and its aims.
Sincerely.
Response 1: Dear Reviewer ,
We appreciate the comments. We've increased the introductory arguments and further clarified the objectives. We increased the discussion, but it was not necessary to increase the number of references because we can get all our arguments from the 25 articles that we list in the references section.
